# Reinforcement Learning with Euclidean Data Augmentation for State-Based Continuous Control

**Jinzhu Luo**
University of South Carolina
jinzhu@email.sc.edu

**Dingyang Chen**
University of South Carolina
dingyang@email.sc.edu

**Qi Zhang**
University of South Carolina
qz5@cse.sc.edu

## Abstract

Data augmentation creates new data points by transforming the original ones for a reinforcement learning (RL) agent to learn from, which has been shown to be effective for the objective of improving the data efficiency of RL for continuous control. Prior work towards this objective has been largely restricted to perturbation-based data augmentation where new data points are created by perturbing the original ones, which has been impressively effective for tasks where the RL agent observes control states as images with perturbations including random cropping, shifting, etc. This work focuses on state-based control, where the RL agent can directly observe raw kinematic and task features, and considers an alternative data augmentation applied to these features based on Euclidean symmetries under transformations like rotations. We show that the default state features used in exiting benchmark tasks that are based on joint configurations are not amenable to Euclidean transformations. We therefore advocate using state features based on configurations of the limbs (i.e., the rigid bodies connected by the joints) that instead provide rich augmented data under Euclidean transformations. With minimal hyperparameter tuning, we show this new Euclidean data augmentation strategy significantly improves both data efficiency and asymptotic performance of RL on a wide range of continuous control tasks. Our code is available on GitHub[1].

## 1 Introduction

While reinforcement learning (RL) has enjoyed impressive success on continuous control problems, especially when empowered by expressive function approximators like deep neural networks, improving its notoriously poor data efficiency remains challenging. Recently, exploiting the idea of data augmentation, the RL community has made significant progress on improving data efficiency as well as the asymptotic performance of RL for continuous control with the agent observing images as the state representations [1, 2, 3, 4]. CURL [1] utilized data augmentation for learning contrastive representations [5, 6] out of their image encoder, as any auxiliary learning objective in the RL setting. Other works, such as DrQ [3, 4] and RAD [2], directly used image data augmentations for RL without any auxiliary objectives. Despite these technical differences in how data augmentation is integrated into RL, these works share the key procedure of perturbation-based image augmentation: to create new data for learning, the image in a state representation is transformed by applying perturbations such as random crop, random shift, color jitter, etc.

Instead of image-based continuous control, this paper focuses on the more primitive problem of *state-based* continuous control, where the agent can observe raw physical quantities such as position and velocity as the features to represent the underlying world state. Like the image-based case, existing work on state-based data augmentation has been relying on introducing perturbations to original state

---

[1] https://github.com/JinzhuLuo/EuclideanDA

38th Conference on Neural Information Processing Systems (NeurIPS 2024).

features, examples including adding Gaussian noise and scaling the features with random amplitudes [2]. However, unlike the image-based case, the perturbed data for state-based continuous control only leads to limited improvements, sometimes even negative effects, on the data efficiency of RL, as found in existing work [2] and our experiments in Section 5. Such ineffectiveness is due to the fundamental difference between image-based and state-based features: from a control-theoretic perspective, the perturbed physical quantities as state-based features are largely uncorrelated with the original ones, in the sense that although the original transition comes from the ground truth dynamics and reward functions, the perturbed transition does not necessarily; this is in contrastive difference from images, where perturbations such as shifting create new views of the same underlying world state.

To overcome the limitations of perturbation-based data augmentation, this paper advocates the alternative of *Euclidean data augmentation* for state-based continuous control. The key idea is that, because our control tasks operate in the 2D/3D space, they exhibit Euclidean symmetries: transformations such as rotation and translation leave the transition dynamics and reward function invariant. Due to the invariancy, these symmetry-preserving Euclidean transformations are ideal operations for data augmentation, because the transformed data will be guaranteed to be valid samples from the task's dynamics and reward.

While there exists prior work on Euclidean symmetries in RL [7, 8, 9, 10], the idea of exploiting them as a data augmentation method for state-based continuous control is surprisingly underexplored. This is because current benchmarks [11, 12] by default use the joint configurations of the robot-like agent as its state features, which are mainly angular quantities that are invariant to Euclidean transformations and therefore not amenable to data augmentation, as we will explain in detail in Section 4. To overcome this, we innovatively use configurations of the limbs, i.e., the rigid body links connected by the joints, as an equivalent state representation to replace the default joint-based one. Because limb configurations are specified by physical quantities such as linear position and velocity, they are equivariant under Euclidean transformations and therefore provide rich augmented data.

Our algorithmic innovations based on the introduced ideas lead to significant improvement in data efficiency and asymptotic performance of RL for state-based continuous control. Building from DDPG [13] as the base RL algorithm, our limb-based state representation alone improves the performance on most tasks from the DeepMind Control Suite [12], a standard continuous control benchmark for RL, and additional Euclidean data augmentation is necessary to obtain the best performance for almost all tasks, especially for the ones that have large degrees of freedom and are historically hard to solve. To name a few hardest tasks, on the Humanoid_run task, standard DDPG achieves an episode reward below 100, while our method achieves 150, both after 5M timesteps; on Hopper3D_hop, standard DDPG achieves an episode reward below 40, while our method achieves more than 200, both after 2M timesteps.

## 2   Related work

**Data augmentation for image-based continuous control.** Existing works in data augmentation for RL mostly have focused on the online setting for image-based continuous control tasks. This is because it is relatively straightforward to obtain augmented images (e.g., through random cropping/shifting), which enables learning representations of the high-dimensional image input that facilitates optimizing the task reward. In this line of work, CURL [1] utilizes contrastive representation learning to the image representations, jointly with the RL objective of reward optimization; RAD [2] and DrQ [3, 4] use the augmented data directly for RL without any auxiliary objective. SVEA [14] focuses on improving stability and sample efficiency under image-based data augmentation by mitigating the high-variance Q-targets. CoIT [15] works on maintaining the invariant information unchanged under image-based data augmentation. Different tasks often benefit from different types of data augmentation, but the manual selection is not scalable. The work of [16] automatically applies the types of data augmentations that are most suitable for the specific tasks.

**Data augmentation for state-based continuous control.** This paper focuses on data augmentation for state-based continuous control, which only a few prior works have explored. Specifically, RAD explores the augmentation of injecting noise to state variables through additive Gaussian or random amplitude scaling [2]. Different from these unprincipled, perturbation-based augmentation, this paper advocates a principled augmentation method through Euclidean symmetries for state-based continuous control. Corrado et al. [17, 18] also consider principled data augmentation transformations that are not

perturbation-based but focus on better leveraging existing transformations, drawing their conclusions mostly from robot navigation and manipulation tasks, while we propose a novel transformation for robot locomotion. Pitis et al. [19, 20] propose a data augmentation transformation that requires local (causal) independence, so that augmentation can be performed via stitching independent trajectories from decomposed, independent parts, which is useful for tasks such as particles moving and two-arm robots with a static base. We instead focus on locomotion tasks that do not exhibit sparse kinematic interactions between limbs and therefore cannot benefit much from their method. For example, the legs of a cheetah-like robot are connected through the moveable torso, which we cannot decompose and stitch their separate trajectories.

**Euclidean symmetries in RL.** Symmetries in RL refer to the structural property of invariant transition dynamics and reward functions under certain transformations, leading to the existence of invariant optimal values and policies [7, 8]. Euclidean symmetries thus correspond to the transformations being translations, rotations, and/or reflections, which are applicable when the state/action space is Euclidean. Most prior works focus on image-based control problems, e.g., Atari games and robot manipulation from image observations [9], or gridworld-like domains exhibiting only discrete symmetries (e.g., 4-way Cartesian rotation) [8, 21]. A closely related work by Abdolhosseini et al. [22] uses reflection-based data augmentation (they call them "mirror") for locomotion. They perform trajectory-level transformations for on-policy RL, which is technically not sound and does not yield significant improvement over the no-augmentation baseline (see Figure 3 therein). Instead of relying on data augmentation, some of these works employ equivariant neural networks as their agent architectures to exploit the symmetries [8, 9], which often incurs significantly higher computational cost. The reason that existing works largely overlook data augmentation based on Euclidean symmetries in state-based continuous control is two-fold: 1) most standard benchmark tasks operate in the 2D space [11, 12], and these planar tasks exhibit limited Euclidean symmetries compared to the 3D counterparts; and 2) the benchmark tasks employ the state representations based on the joints' configurations and velocities, which in nature exhibit very limited Euclidean symmetries. This paper addresses these issues to unlock the potential of data augmentation through Euclidean symmetries for state-based control via RL.

## 3 Preliminaries

### 3.1 Online RL for state-based continuous control

We formulate a state-based continuous control task (or, simply, task) as a Markov decision process (MDP), defined as a tuple $(\mathcal{S}, \mathcal{A}, p, r, \gamma)$. At each discrete time step $t$, the agent fully observes the state in the continuous state space, $s_t \in \mathcal{S} \subset \mathbb{R}^d$, and chooses an action in the continuous action space to take, $a_t \in \mathcal{A} \subset \mathbb{R}^m$; the next state is sampled from the transition dynamics $p$ with conditional probability density $p(s_{t+1}|s_t, a_t)$, while the reward function $r : \mathcal{S} \times \mathcal{A} \to \mathbb{R}$ yields a scalar feedback $r_t := r(s_t, a_t)$. The agent uses a policy $\pi : \mathcal{S} \to \Delta(\mathcal{A})$ to choose actions where $\Delta(\mathcal{A})$ is the collection of probability measures on $\mathcal{A}$, yielding a trajectory of states, actions, and rewards, $(s_0, a_0, r_0, s_1, \ldots)$, where the initial state $s_0$ is sample from an initial state distribution with density $d_0$ and $a_t \sim \pi(\cdot|s_t)$ Without prior knowledge of $p$ and $r$, the reinforcement learning (RL) agent's goal is to obtain a policy from its experience (i.e., yielded trajectories) which maximizes the expected cumulative discounted reward, $\mathbb{E}_\pi[\sum_{t=0}^\infty \gamma^t r_t]$ where $\gamma \in [0, 1)$. The online RL setting assumes the agent has the ability to interact through the MDP to obtain an increasing amount of trajectories while refining its policy.

### 3.2 Deep Deterministic Policy Gradient

Deep Deterministic Policy Gradient (DDPG) [13] is a state-of-the-art online RL algorithm for continuous control, which parameterizes and learns a critic $Q_\theta : \mathcal{S} \times \mathcal{A} \to \mathbb{R}$ and a deterministic policy $\pi_\phi : \mathcal{S} \to \mathcal{A}$. The critic is trained by minimizing the one-step temporal difference (TD) error $\mathcal{L}(\theta) = \mathbb{E}_{(s_t, a_t, r_t, s_{t+1}) \sim \mathcal{D}_{\text{online}}}[(Q_\theta(s_t, a_t) - r_t - \gamma Q_{\bar{\theta}}(s_{t+1}, \pi_\phi(s_{t+1})))^2]$, where $\mathcal{D}_{\text{online}}$ is the replay buffer storing past trajectories and $\bar{\theta}$ is an exponential moving average of the critic parameter, referred to as the target critic. Concurrently, the policy is trained by employing Deterministic Policy Gradient (DPG) [23] and minimizing $\mathcal{L}(\phi) = \mathbb{E}_{s_t \sim \mathcal{D}_{\text{online}}}[-Q_\theta(s_t, \pi_\phi(s_t))]$, so that $\pi_\phi(s_t)$ approximates $\arg\max_a Q_\theta(s_t, a)$. The policy, after adding noise for exploration, is used as a data-collection policy

to generate and store new trajectories in replay buffer $\mathcal{D}_{\text{online}}$, which is interleaved with the critic and the actor parameter update steps.

### 3.3 Perturbation- and symmetry-based data augmentation in RL

**Perturbation-based data augmentation.** Data augmentation generates new, artificial data through transformations of original data. Better data efficiency can often be achieved by learning from both the original and augmented data. For mastering continuous control tasks via RL, prior works perform data augmentation by perturbation, through 1) transformations of adding noise to original data, examples including performing random image cropping, color jittering, translations, and rotations when states can only be observed and represented as images (i.e., image-based continuous control) [24, 25, 1, 2, 3, 4] and 2) transformations of adding Gaussian noise and random amplitude scaling when states (e.g., positions and velocities) can be directly observed by the agent [2].

**Symmetry-based data augmentation.** This work focuses on state-based continuous control and departs from prior works by performing data augmentation through *MDP symmetries* [7, 8]. Symmetries in an MDP, if existing, can be described by a set of transformations on the state-action space indexed by $g \in G$, where such a transformation leaves the transition dynamics and reward function invariant. Formally, we define a state transformation and a state-dependent action transformation as $T_g^{\mathcal{S}} : \mathcal{S} \to \mathcal{S}$ and $T_{g,s}^{\mathcal{A}} : \mathcal{A} \to \mathcal{A}$, respectively, and the invariance is expressed accordingly as, for all $g \in G, s, s' \in \mathcal{S}, a \in \mathcal{A}$,

$$p(s'|s,a) = p\left(T_g^{\mathcal{S}}[s'] \mid T_g^{\mathcal{S}}[s], T_{g,s}^{\mathcal{A}}[a]\right) \quad \text{and} \quad r(s,a) = r\left(T_g^{\mathcal{S}}[s], T_{g,s}^{\mathcal{A}}[a]\right). \tag{1}$$

The invariance naturally makes these transformations sound and effective for data augmentation purposes, especially in continuous control tasks where transition dynamics is often (near-)deterministic.

Next, we detail our data augmentation method based on Euclidean symmetries for continuous control.

## 4 Our method: Euclidean data augmentation for continuous control

### 4.1 Kinematics of our continuous control tasks

We consider the continuous control problem for an agent (e.g., a robot) consisting of $n$ rigid bodies, referred to as *limbs*, that are connected by *joints* to form a certain morphology and can move in the 2D planar space or the 3D space. Examples of such agents include 2D Cheetah and 2D Hopper from DeepMind Control Suite (DMControl) [12] as well as their 3D variants [26], which will be presented in our experiments. Without loss of generality, we present our method in the 3D setting.

**Tree-structured morphology and actuation.** The connected limbs form a certain morphology that can be represented as an undirected graph, where each node $i \in \{1, \cdots, n\}$ represents a limb and an undirected edge $(i, j)$ exists if there is a joint connecting nodes/limbs $i$ and $j$. We focus on the case where the graph is connected and acyclic to become a tree, where the root node, indexed as $i = 1$, is often the torso or the base in the agent's morphology. Any two connected limbs are rigidly attached to each other via their joint to provide degrees of freedom (DoFs) between the two limbs. A joint provides either 1-, 2-, or 3-DoF rotation for the child node/limb relative to its parent: in general, as a 3D rigid body, the child can rotate about yaw-, pitch-, and/or roll-axes relative to its parent, creating 3 DoFs; for a 2D planer task, the agent operates in the $xz$-plane and the joints therein can only rotate about the $y$-axis (pitch), creating only 1 DoF. The joints are actuated to produce a scalar torque for each of these rotation DoFs. We let $d_i \in \{1, 2, 3\}$ denote the number of DoFs of the joint with node $i > 1$ as its child, and the collection of $d_i$ scalar torques is denote as $\boldsymbol{a}_i \in \mathbb{R}^{d_i}$, resulting in the action for the agent as $\boldsymbol{a} = (\boldsymbol{a}_i)_{i>1} \in \mathbb{R}^m$ where $m = \sum_{i>1} d_i$. The root node, as a 3D rigid body, has up to 6 DoFs relative to the global frame, and we let $d_1 \leq 6$ denote the number of its DoFs.

**Kinematic features.** As the action of torques specifies acceleration, a Markovian state of such an agent has to specify its configuration, i.e., how the limbs occupy the space and how the joints are rotated, as well as its velocity, i.e., the time derivative of its configuration. We refer to these physical quantities as the agent's *kinematic features* and enumerate here several of them that will be closely related to our discussion:

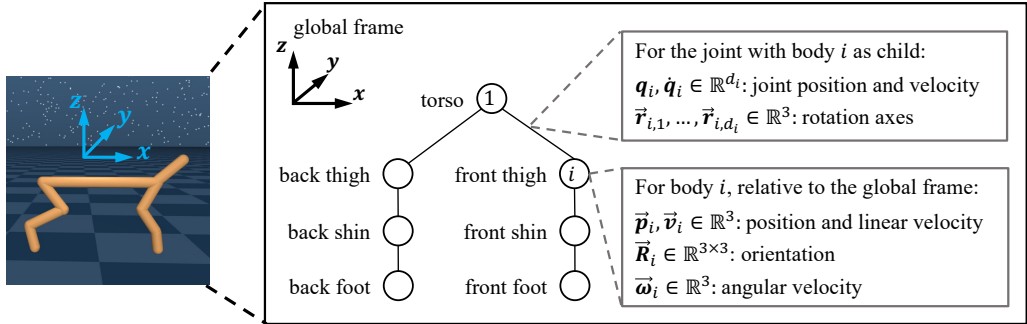

Figure 1: Illustration of Cheetah from DMControl, including its rendering, tree-structured morphology with the nodes being the limbs and the edges the joints, and the state features.

- $q_i, \dot{q}_i \in \mathbb{R}^{d_i}$ respectively denote the configuration and the velocity (i.e., the configuration's time derivative) of the joint with limb $i$ as its child. For $i > 1$, we have $d_i \leq 3$ corresponding to each active joint rotation axis, and $q_i$ consists of the rotation angles. For $i = 1$, there is no joint with limb $i = 1$ (i.e., torso) as its child; in this case, $q_1 \in \mathbb{R}^{d_1}$ specifies the configuration corresponding to all $d_1 \leq 6$ DoFs of the limb.

- $\vec{r}_{i,1}, \cdots, \vec{r}_{i,d_i} \in \mathbb{R}^3$ denote the $d_i$ rotation axes of the joint with limb $i > 1$ as its child, relative to the global frame.

- $\vec{p}_i, \vec{v}_i \in \mathbb{R}^3$ respectively denote the (translational) position and linear velocity (i.e., the position's time derivative) of limb $i \geq 1$, relative to the global frame.

- $\vec{R}_i \in \mathbb{R}^{3\times3}$ and $\vec{\omega}_i \in \mathbb{R}^3$ respectively denote the Cartesian orientation and angular velocity (i.e., the orientation's time derivative) of limb $i \geq 1$, relative to the global frame.

**Task features.** Our continuous control tasks are mostly of locomotion, i.e., the agent is tasked to move (part of) itself from one place to another. In tasks where the agent is tasked to move in a certain direction (e.g., Cheetah), we let $\vec{t} \in \mathbb{R}^3$ denote the unit vector in that direction. In tasks where the agent aims to reach a target point $\vec{p}_* \in \mathbb{R}^3$ with one of its limb $i$ (e.g., Reacher where $i = n = 3$ being the last limb), we let $\vec{t} = \vec{p}_* - \vec{p}_i \in \mathbb{R}^3$ denote the vector from the limb to the target point.

**Example: 2D/3D Cheetah.** The original Cheetah from DMControl operates in the $xz$-plane [12] with the target direction being $\vec{t} = [1\ 0\ 0]$, i.e., the agent is tasked to move along the positive $x$-axis.

Its torso is the root node and has $d_1 = 3$ DoFs: moving along the $x$-axis with position $p_1^x \in \mathbb{R}$, moving along the $z$-axis with position $p_1^z \in \mathbb{R}$, and rotating about the $y$-axis with rotation angle $\beta_1^y \in [0, 2\pi)$, and correspondingly we have $q_1 = (p_1^x, p_1^z, \beta_1^y) \in \mathbb{R}^3$. The other 6 limbs (front/back thigh, shin, and foot) form the front and back legs attached to the torso, forming the morphology tree as shown in Figure 1. For the 2D planar case, the 6 joints can only rotate about the $y$-axis, and therefore, for $i \in \{2, \cdots, 7\}$, we have $d_i = 1$, $q_i = (\beta_i^y) \in \mathbb{R}^1$ with $\beta_i^y$ being the rotation angle, and its only rotation axis $\vec{r}_{i,1}$ is fixed along the $y$-axis. In our experiments, we extend this 2D Cheetah to its 3D variant, where the torso has all $d_1 = 6$ DoFs to freely move the in 3D space and each of the 6 joints, $i \in \{2, \cdots, 7\}$, has $d_i = 3$ DoFs to be able to freely rotate about the yaw-, pitch-, and roll- axes $(\vec{r}_{i,1}, \vec{r}_{i,2}, \vec{r}_{i,3})$.

### 4.2 $\mathrm{SO}_{\vec{g}}(3)$-data augmentation

If the agent is placed in the free space (free of obstacles and external forces), its transition dynamics and reward function are invariant to the transformations including translations, rotations, and reflections. Formally, these transformations together form the Euclidean group of $G = \mathrm{E}(3)$ in Equation (1), where the state should include the agent's kinematic features (i.e., configuration and

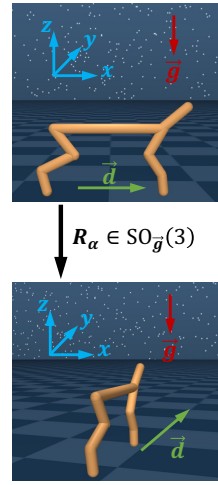

Figure 2: $\mathrm{SO}_{\vec{g}}(3)$ rotation in Cheetah.

velocity) and the task features. This is the basis of our method of Euclidean data augmentation.

For the tasks in our experiments, the agent is not placed in the free space and, instead, is placed on the level surface of $xy$-plane (i.e., the ground) and subject to the external force of gravity $\vec{g} \in \mathbb{R}^3$ towards the negative $z^-$-axis. Therefore, the task is not invariant to arbitrary rotations/reflections but only to rotations about the direction of $\vec{g}$ (or equivalently, the $z$-axis), which subsumes reflections along this direction. For the reasons mentioned above, our Euclidean data augmentation only includes rotations about $\vec{g}$, which form a subgroup of $\mathrm{SO}(3)$ (which includes all 3D rotations) and is denoted with its matrix representation as $\mathrm{SO}_{\vec{g}}(3) = \{\boldsymbol{R} \in \mathbb{R}^{3 \times 3} \mid \boldsymbol{R}^\top \boldsymbol{R} = \boldsymbol{I}, \det \boldsymbol{R} = 1, \boldsymbol{R}\vec{g} = \vec{g}\}$. Alternatively, the matrices can be parameterized by the angle rotated about $z$-axis (i.e., yaw) as $\alpha \in [0, 2\pi)$ as $\mathrm{SO}_{\vec{g}}(3) = \{\boldsymbol{R}_{\alpha,\beta,\gamma} \in \mathbb{R}^{3 \times 3} \mid \alpha \in [0, 2\pi), \beta = \gamma = 0)\}$ where $\alpha$, $\beta$, and $\gamma$ are yaw, pitch, and roll angles, respectively. We will abbreviate the matrix as $\boldsymbol{R}_\alpha$ since $\beta = \gamma = 0$.

Note that other Euclidean transformations, i.e., translations or reflections, are not available in our tasks: As explained later, the agents in the tasks use egocentric representation and therefore it's already translation-invariant. Reflections require gait symmetry (e.g., the left leg has the same length, stiffness, etc.), which the agents in the tasks do not necessarily have. Some prior work has modified the agent to enforce gait symmetry before applying reflections (e.g., Corrado and Hanna [18]), but we do not.

For a state rotated by $\boldsymbol{R}_\alpha$, the kinematic features and task features are accordingly transformed in the following three ways (see Figure 2 for an illustration in Cheetah):

- *Equivariant features*: We have denoted these features with arrow as 3D vector $\vec{x} \in \mathbb{R}^3$ which will be rotated into $\boldsymbol{R}_\alpha \vec{x} \in \mathbb{R}^3$, including $\vec{r}_{i,1}, \cdots, \vec{r}_{i,d_i}, \vec{p}_i, \vec{v}_i$, columns of $\vec{\boldsymbol{R}}_i, \vec{\omega}_i$, and $\vec{t}_i$.

- *Invariant features*: For the joint with a non-torso limb $i > 1$ as its child, its configuration $\boldsymbol{q}_i$ specifies the rotation angles to the active rotation axes. Therefore, $\{\boldsymbol{q}_i, \dot{\boldsymbol{q}}_i\}_{i>1}$ and torque action $\boldsymbol{a}$ will stay invariant.

- *Transformation of $\boldsymbol{q}_1$ and $\dot{\boldsymbol{q}}_1$*: Note that elements of the torso's position $\vec{p}_1$ are also part of $\boldsymbol{q}_1$. This part of $\boldsymbol{q}_1$ will be transformed accordingly. The rest of $\boldsymbol{q}_1$ includes $\beta_1^z, \beta_1^y$, and/or $\beta_1^x$, the yaw, pitch, and/or roll angles of the torso. Since rotation $\boldsymbol{R}_\alpha$ is a yaw-only rotation, we have $\beta_1^z \to \beta_1^z + \alpha$ while $\beta_1^y$ and $\beta_1^x$ will stay invariant. Similar arguments apply to $\dot{\boldsymbol{q}}_1$.

We below present a few implementation details of $\mathrm{SO}_{\vec{g}}(3)$-data augmentation in our experiments.

**Limb-based kinematic state representation.** Besides the task features, the Markovian state should include, for the agent itself, kinematics features that are sufficient to represent its configuration and velocity. Not all kinematic features introduced in Section 4.1 are independent and therefore a sufficient kinematic state representation needs only to include a subset of them. Current benchmarks for RL-based continuous control (e.g., DMControl [12] and Gym-MuJoCo [11]) all employ the *joint-based* kinematic state representation, which includes only joint configurations and velocities $\{\boldsymbol{q}_i, \dot{\boldsymbol{q}}_i\}_{i \geq 1}$, which is indeed a valid kinematic state representation because all limbs are rigid bodies. However, as we have discussed, this joint-based kinematic state representation is not amenable to $\mathrm{SO}_{\vec{g}}(3)$-data augmentation, since most of its features are invariant (only $\boldsymbol{q}_1$ and $\dot{\boldsymbol{q}}_1$ are not invariant).

To provide rich augmented data under $\mathrm{SO}_{\vec{g}}(3)$, we use an alternative kinematic state representation that is mostly based on limb features, including $\vec{\boldsymbol{R}}_1, \vec{\omega}_1, \{\vec{p}_i, \vec{v}_i\}_{i \geq 1}$, and $\{\vec{r}_{i,1}, \cdots, \vec{r}_{i,d_i}\}_{i>1}$. For torso $i = 1$, its configuration and velocity are included in $\{\vec{\boldsymbol{R}}_1, \vec{\omega}_1, \vec{p}_1, \vec{v}_1\}$. For limb $i > 1$, its translational position and velocity are included in $\{\vec{p}_i, \vec{v}_i\}$; its orientation is included in its rotation axes $\{\vec{r}_{i,1}, \cdots, \vec{r}_{i,d_i}\}$, so we do not further include $\vec{\boldsymbol{R}}_i$; and its angular velocity can be recovered from its rotation axes and its velocity, so we do not further include $\vec{\omega}_i$.

The kinematic features listed above are both sufficient and necessary in general for an agent operating in the 3D space. For a 1-DoF joint ($d_i = 1$) that is typical in 2D planar tasks, the orientation of the torso, $\vec{\boldsymbol{R}}_1$, is sufficient to determine the orientation of its active rotation axis. Therefore, we exclude the rotation axes for all the 1-DoF joints and include only for the 2- and 3-DoF joints, resulting in the final set of features in our limb-based kinematic state representation:

$$\vec{\boldsymbol{R}}_1, \vec{\omega}_1, \{\vec{p}_i, \vec{v}_i\}_{i \geq 1}, \{\vec{r}_{i,1}, \cdots, \vec{r}_{i,d_i} : i > 1, d_i > 1\}. \tag{2}$$

**Egocentric kinematic features.**    In these benchmark continuous control tasks, the agent can only observe its egocentric position, that is, it observes the position of its torso as the origin and the positions of other limbs and the target relative to its torso. For example, when using the joint-based kinematic state representation for 2D Cheetah, the configuration of torso $\boldsymbol{q}_1 = (p_1^x, p_1^z, \beta_1^y) \in \mathbb{R}^3$ is observed as $(0, p_1^z, \beta_1^y)$. Consistently, our limb-based representation subtracts the torso's translational position for every limb, i.e., $\vec{\boldsymbol{p}}_i \to \vec{\boldsymbol{p}}_i - \vec{\boldsymbol{p}}_1$ for all limbs $i \geq 1$.

**Additional sensory observations.**    Besides the kinematic and task features, the agent has additional sensory observations in its original state representation that can be classified as either equivariant or invariant features under $\mathrm{SO}_{\vec{\boldsymbol{g}}}(3)$ rotations. For example, Hopper from DMControl has two sensors for detecting the touch of its two feet, each yielding an invariant scalar feature. Refer to Table 2 in Appendix A.1 for a complete list of sensory observations for all tasks. Our complete state representation includes these sensory inputs, along with the kinematic and task features, to ensure equivalence to the original state representation.

## 4.3   Integration with DDPG

Our data augmentation method can be applied to any off-policy RL algorithm, and our empirical study applies it to DDPG as follows. After sampling a batch of $B$ transitions of the form $(s_t, a_t, r_t, s_{t+1})$ from online replay buffer $\mathcal{D}_{\mathrm{online}}$, we randomly choose a subset of $B_{\mathrm{aug}} (\leq B)$ transitions and perform random rotations $\boldsymbol{R}_\alpha \in \mathrm{SO}_{\vec{\boldsymbol{g}}}(3)$ to them, with $\alpha$ randomly chosen for each of the $B_{\mathrm{aug}}$ transitions. $B_{\mathrm{aug}}$ is the only additional hyperparameter that our method introduces on top of the original DDPG. Specifically, we transform states $s_t$ and $s_{t+1}$ as described in Section 4.2, keep action $a_t$ and reward $r_t$ invariant, and use the transformed transition to compute the DDPG losses (cf. Section 3.2).

## 5   Experiments

**Environment and tasks.**    All the tasks in our experiments are provided by the DeepMind Control Suite (DMControl) [12] powered by the physics simulator of MuJoCo [27], which has become a common benchmark for RL-based continuous control. Specifically, we include 7 tasks originally from DMControl: Cheetah_run, Hopper_hop, Walker_run, Quadruped_run, Reacher_hard, Humanoid_run, Humanoid_stand, and our modified Cheetah3D_run, Hopper3D_hop, and Walker3D_run, which are the 3D variants of their original 2D planar counterparts. The original tasks involving Quadruped and Humanoid are already 3D, while it is not straightforward to extend Reacher_hard to a 3D variant. This results in a total number of 10 tasks. For all tasks in DMControl, an episode corresponds to 1000 steps, where a per-step reward is in the unit interval, i.e., $r_t \in [0, 1]$. The original tasks in DMControl employ the joint-based kinematic state representation. We use MuJoCo's built-in data structures and functions (e.g., `mjData.xpos` for the translational position, `mjData.object_velocity` for linear and angular velocity [28]) to obtain the kinematics features needed for our limb-based representation. Refer to Appendix A.1 for a detailed description of the state features for all the tasks.

**Baselines.**    We build our data augmentation on top of **DDPG**, following the implementation in [10] that has achieved state-of-the-art performance on the chosen tasks (comparable or better than, e.g., TD3 [29], Soft Actor-Critic (SAC) [30], e.g., refer to [10] for a comparison). Refer to Appendix A.2 for a full list of the DDPG's hyperparameters. For reference, we also run and provide results of standard **SAC**. Our method introduces only one additional hyperparameter, $B_{\mathrm{aug}}$, the number of transitions to be transformed in a batch of $B$ transitions. We perform a hyperparameter search over $B_{\mathrm{aug}}/B =: \rho_{\mathrm{aug}} \in \{0, 25\%, 50\%, 75\%, 100\%\}$ separately for each task, while keeping all other hyperparameters the same as DDPG default. We compare our $\mathrm{SO}_{\vec{\boldsymbol{g}}}(3)$-data augmentation with two alternatives considered in the prior work of [2], both operating on the original joint-based kinematic state representation: The Gaussian noise (**GN**) method adds a standard Gaussian random variable to the state vector, i.e., $s \to s + z$ where $z \sim \mathcal{N}(0, I)$; and random amplitude scaling (**RAS**) multiplies the uniform random variable to the state vector element-wise, i.e., $s \to s \odot z$ with $\odot$ denoting element-wise product, where $z \sim \mathrm{Uni}[\alpha, \beta]$ with $\alpha = 0.5, \beta = 1.0$ as chosen in the prior work of [2]. We additionally compare to using equivariant neural networks, specifically **SEGNN** [31], to instantiate the agent's policy and critic, as an alternative method to exploit the Euclidean symmetries. Refer to Appendix A.3 for the details of our SEGNN implementation.

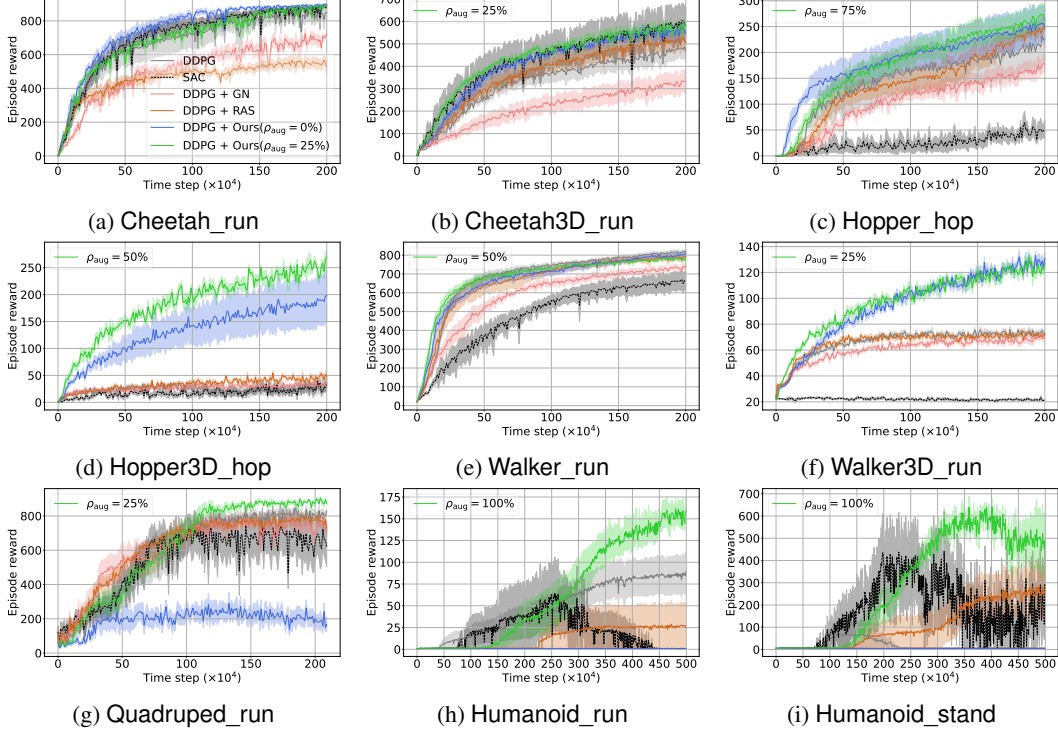

Figure 3: Learning curves comparing data efficiency of our method again all baselines except for SEGNN on 9 out of the 10 tasks, excluding the task of Reacher_hard. The results involving SEGNN and Reacher_hard are deferred to Figure 4.

**Training details.** Each training run consumes 2M time steps (1000 steps per episode for 2000 episodes) for all the tasks except for Humanoid_run and Humanoid_stand, for which each training run consumes 5M steps. Evaluation is performed every 10000 steps by averaging the episodic rewards of 10 episodes, which is reported in our figures as the y-axis. In all the figures, we plot the mean performance over 5 seeds together with the shaded regions which represent 95% confidence intervals. The training runs are computed by NVIDIA V100 single-GPU machines, each taking roughly 2 hours to finish 1M training time steps for our method and all the baselines except for the SEGNN baseline, which takes roughly 70 hours to finish 1M steps.

## 5.1 Comparison with standard RL and prior data augmentation methods

As SEGNN is prohibitively expensive in computation, we only run it on the task of Reacher_hard, the smallest one among all 10 tasks, and defer the results to Section 5.2. Figure 3 presents the learning curves to compare our method against all other baselines on the rest 9 tasks. For our method, we present both $\rho_{\mathrm{aug}} = 0\%$ and the task-specific best positive $\rho_{\mathrm{aug}} \in \{25\%, 50\%, 75\%, 100\%\}$ to separate the effects of our limb-based kinematic state representation and the data augmentation on top of it. We make the following observations from the results:

(i) DDPG is comparable to or significantly better than SAC in all 9 tasks except for Cheetah_run and Humanoid_run, which justifies our choice of DDPG as the base RL algorithm.

(ii) As existing alternative data augmentation methods for state-based continuous control, the baselines of GN and RAS do not offer significant improvement in data efficiency. They even introduce negative effects on performance on most tasks, except for Cheetah3D_run and Humanoid_stand where RAS improves data efficiency over its based algorithm of DDPG.

(iii) With $\rho_{\mathrm{aug}} = 0\%$, i.e., with the limb-based state representation alone, our method improves data efficiency over DDPG on the 6 tasks shown in the first 2 rows in Figure 3, and in 5 out of the 6 tasks (excluding Hopper3D_hop), $\rho_{\mathrm{aug}} = 0\%$ is comparable to the best $\rho_{\mathrm{aug}} > 0\%$. On the rest of

4 tasks (Hopper3D_hop and 3 tasks in the last row), a positive $\rho_{\mathrm{aug}} > 0$ is crucial to attain the best performance among all the methods.

To summarize, our method reliably improves DDPG, the state-of-the-art RL algorithms on the continuous control tasks. The improvements are remarkably significant, especially on tasks with rich 3D DoFs and/or large numbers of state features: for example, the improvements on Hopper3D_run are more salient on its 2D variant Hopper_run; our aggressive data augmentation ($\rho_{\mathrm{aug}} = 100\%$) is necessary to achieve the best performance on Humanoid_run and Humanoid_stand, the hardest two tasks where no baseline is able to learn on both.

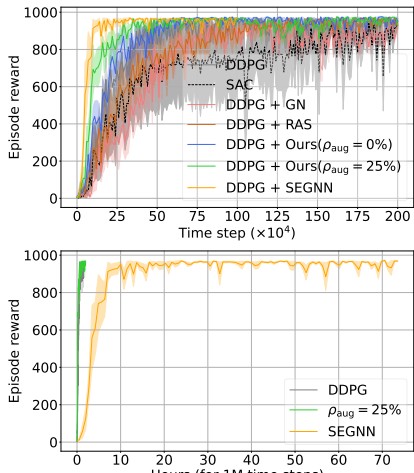

## 5.2 Comparison with equivariant agent architecture

Employing equivariant neural networks such as SEGNN as the agent policy and critic architecture is an alternative method to exploit the Euclidean symmetry but at the same time more computationally expensive than data augmentation. As we are not able to afford to finish the SEGNN baseline for all tasks, we have run it only for Reacher_hard, the smallest one by the number of limbs/joints. Figure 4 reports the learning curves of our method, SEGNN, and all other baselines on Reacher_hard, with the x-axis being time steps and hours to compare data and computational efficiency, respectively. As the results show, SEGNN is the most data efficient, followed by our data augmentation method ($\rho_{\mathrm{aug}} = 25\%$), both clearly outperforming the others. However, our method introduces minimal computation in addition to its DDPG base algorithm, both taking roughly 2 hours to finish 1M steps after convergence, while SEGNN takes more than 70 hours to finish 1M steps and more than 10 hours to converge.

Figure 4: Learning curves of data efficiency (*top*) and run time for 1M steps in total (*bottom*) for our method and all baselines on Reacher_hard.

## 5.3 Effect of $\rho_{\mathrm{aug}}$

In Sections 5.1 and 5.2, we have presented the learning curves of our method with $\rho_{\mathrm{aug}} = 100\%$ and best task-specific $\rho_{\mathrm{aug}} > 0\%$. Here, we present further results detailing the effect of $\rho_{\mathrm{aug}}$, with the learning curves of all values of $\rho_{\mathrm{aug}}$ on all tasks given in Appendix B.1. We observe that: 1) On relatively simple tasks with 2D planer DoFs and small numbers of limbs/joints, $\rho_{\mathrm{aug}} > 0\%$ does not provide significant improvements over $\rho_{\mathrm{aug}} = 0\%$ examples including Cheetah_run as shown in Figure 5a; 2) Data augmentation ($\rho_{\mathrm{aug}} > 0\%$) is crucial to best performance on hard tasks with 3D DoFs and large numbers of limbs/joints, examples including Hopper3D_hop and Humanoid_run in Figures 5b and 5c, respectively; and 3) The best $\rho_{\mathrm{aug}}$ is task-specific and not necessarily the largest possible $\rho_{\mathrm{aug}} = 100\%$. For example, $\rho_{\mathrm{aug}} = 100\%$ hinders performance on Hopper3D_hop but is the best on Humanoid_run.

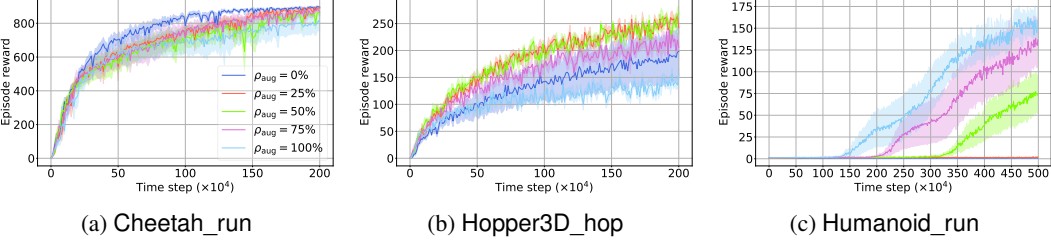

(a) Cheetah_run      (b) Hopper3D_hop      (c) Humanoid_run

Figure 5: Learning curves of our method on the effect of $\rho_{\mathrm{aug}}$ on three sample tasks.

These observations underscore the task-specific nature of augmentation strategies, which aligns with findings from previous studies in the field of computer vision, where the idea of data augmentation originated and has prevailed. Cubuk et al. [32] demonstrated that different datasets such as CIFAR-10, SVHN, and ImageNet require distinct augmentation strategies. Similarly, Zoph et al. [33] showed significant differences in the optimal augmentation strategies for various object detection

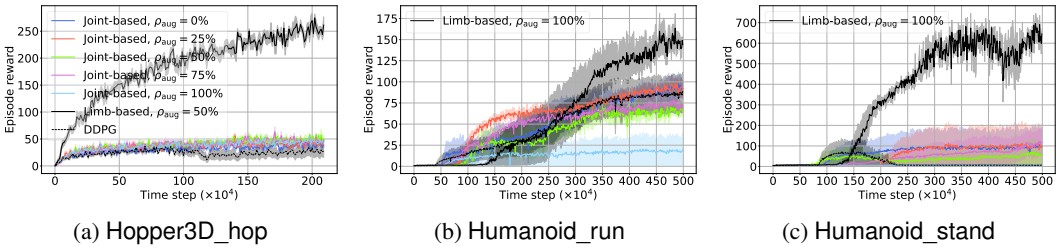

(a) Hopper3D_hop  (b) Humanoid_run  (c) Humanoid_stand

Figure 6: Comparison with joint-based data augmentation on three sample tasks.

tasks, highlighting the task-specific requirements of augmentation policies. Ho et al. [34] further emphasized this point in their study on Population Based Augmentation, demonstrating that the best augmentation policies vary across tasks, underscoring the need to tailor augmentation strategies to specific tasks and datasets.

## 5.4 Comparison with joint-based data augmentation

Indeed, one can perform $\mathrm{SO}_{\bar{g}}(3)$ rotations on the original joint-based state representation. In the presented tasks, only the torso ($i = 1$) has features for orientation (in its up to 6 DoFs) while features of other limbs ($i > 1$) are just angles and angular velocities of the hinges. Therefore, under a rotation, only the torso's orientation features are changed, while other features stay unchanged. Therefore, we hypothesize this data augmentation by rotating original joint-based features would bring little benefit to the tasks. This is because torso features make up only a fraction of all features when the number of limbs is large (which is the case in these tasks). Our limb-based representation instead rotates all limbs to provide richer augmentation. The hypothesis is supported by results of prior work (e.g., see Figure 16 in Corrado and Hanna [18]). We here also conduct a comparison, with results in Figure 6 showing that joint-based data-augmentation is less efficient than our limb-based method.

## 6  Conclusion

We have motivated, formalized, and developed the idea and method of a new data augmentation method that aims to improve the performance of RL, as measured by its data efficiency and asymptotic reward, for state-based continuous control. The key idea of our data augmentation method is to create additional training data by performing Euclidean transformations on the original states, which is justified by the Euclidean symmetries inherent in the continuous control tasks. To make the states more amenable to Euclidean transformations, we turn to a state representation based on limbs' kinematic features, rather than on joints' configurations as done by default in prior works. Our new method significantly improves the performance for a wide range of state-based continuous control tasks, especially for tasks with rich 3D motions and large numbers of limbs and joints.

**Limitations.**  Our work focuses on robot locomotion tasks as instances of continuous control, which clearly exhibit Euclidean symmetries. Other robotics tasks (e.g., navigation) and many applications that operate in the 3D physical space should also exhibit Euclidean symmetries. There are indeed continuous control tasks that might not exhibit Euclidean symmetries, such as those in electrical and power engineering, which we do not consider. We observe two limitations that would inspire future work: 1) To obtain the best performance, our data augmentation method needs task-specific turning of its hyperparameter, $\rho_{\mathrm{aug}}$, that controls the proportion of data to be transformed for learning. Future work in this direction is needed to ease task-specific turning, either by adaptive tuning strategies [35] or training techniques to make learning more robust and less sensitive to the hyperparameter [36]; 2) The proposed method requires knowledge and annotation of strict Euclidean symmetries in the continuous tasks, which might not be easily available especially when the external forces are noisy and hard to detect, e.g., including wind conditions in the wild field along with gravity. This prompts future work of automatic discovery and exploitation of irregular, approximate Euclidean symmetries.

## Acknowledgments and Disclosure of Funding

The authors acknowledge funding support from Qi Zhang's NSF CAREER award 2237963. Any opinions, findings, conclusions, or recommendations expressed here are those of the authors and do not necessarily reflect the views of the sponsors. The authors thank the anonymous reviewers for their insightful and constructive reviews.

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

# A  Implementation details

## A.1  Task details

**Kinematic features.**  Table 1 lists for all the 10 tasks the kinematic parameters including number of limbs $n$, number of DoFs of the torso limb $d_1$, numbers of 1- 2-, and 3-DoF joints $n_{1\text{-DoF}}$, $n_{2\text{-DoF}}$, and $n_{3\text{-DoF}}$. There are three possible values for $d_1$:

- $d_1 = 0$. The torso is fixed, e.g., in Reacher_hard.

- $d_1 = 3$. The torso has all 3 DoFs: moving along the $x$-axis, moving along the $z$-axis, and rotating about the $y$-axis, which is the case in 2D planar tasks.

- $d_1 = 6$. The torso has all 6 DoFs, which is the case in 3D tasks.

These parameters are sufficient to recover other kinematic parameters of a task. For example, action size $m = \sum_{i>1} d_i = n_{1\text{-DoF}} + 2n_{1\text{-DoF}} + 3n_{1\text{-DoF}}$; according to (2), the size of our limb-based kinematic features of a task should be $3 \times 3 + 3 + 3n + 3(2n_{1\text{-DoF}} + 3n_{1\text{-DoF}})$.

A caveat is that, for the number of limbs $n$ in Table 1, we do not count the bodies that do not have any DoF with respect to its parent, examples including the finger with respect to wrist in Reacher_hard. An exception is the torso in Reacher_hard, where we do count it in $n$. Such bodies are welded to its parent. By doing so, we keep the relationship of $n = n_{1\text{-DoF}} + n_{2\text{-DoF}} + n_{3\text{-DoF}} + 1$ in Table 1. In our implementations, however, we do include the kinematic features of these bodies.

Table 1: Kinematic parameters of the tasks. $n$: number of limbs. $d_1$: number of DoFs of the torso limb ($i = 1$). $n_{1\text{-DoF}}, n_{2\text{-DoF}}, n_{3\text{-DoF}}$: number of 1-, 2-, 3-DoF joints.

| Task | $n$ | $d_1$ | $n_{1\text{-DoF}}$ | $n_{2\text{-DoF}}$ | $n_{3\text{-DoF}}$ |
|---|---|---|---|---|---|
| Cheetah_run | 7 | 3 | 6 | 0 | 0 |
| Cheeetah3D_run | 7 | 6 | 0 | 0 | 6 |
| Hopper_hop | 5 | 3 | 4 | 0 | 0 |
| Hopper3D_hop | 5 | 6 | 0 | 0 | 4 |
| Walker_run | 7 | 3 | 6 | 0 | 0 |
| Walker3D_run | 7 | 6 | 0 | 0 | 6 |
| Quadruped_run | 13 | 6 | 8 | 4 | 0 |
| Reacher_hard | 3 | 0 | 2 | 0 | 0 |
| Humanoid_run/stand | 13 | 6 | 5 | 5 | 2 |

**Sensory observations.**  Table 2 lists the sensory observations for all the 10 tasks.

Table 2: Sensory observations in the tasks.

| Task | Sensory Observations | |
|---|---|---|
| | Number of Invariant Features | Number of Equivariant Features |
| Cheetah_run | - | - |
| Cheetah3D_run | - | - |
| Hopper_hop | 2:two foot touch sensors | - |
| Hopper3D_hop | 2:two foot touch sensors | - |
| Walker_run | 1:the height of the torso | - |
| Walker3D_run | 1:the height of the torso | - |
| Quadruped_run | 4: the 4 foot torque | 4: the 4 foot force |
| | 1: the dot-product of the torso z-axis and the global z-axis | 1: the acceleration |
| | 1:the gyroscope | |
| | 4:the 4 egocentric state | |
| Reacher_hard | 1: the vector from target to finger | - |
| Humanoid_run/stand | 1: the z-projection of the torso orientation matrix | 1: the velocity of the center-of-mass |
| | | 4: the 4 end effector positions in egocentric frame |

## A.2 DDPG hyperparameters

Table 3 presents the full list of DDPG hyperparameters used in our method and baselines with DDPG as the base RL algorithm, including standard DDPG, DDPG + GN, DDPG + RAS, and DDPG + Ours. The hyperparameters and other implementation details of the SEGNN-based baseline is deferred to Section A.3.

Table 3: DDPG hyperparameters used in our experiments.

| Hyperparameter | Setting |
|---|---|
| Learning rate | $1e{-}4$ |
| Optimizer | Adam |
| $n$-step return | 3 |
| Mini-batch size | 256 |
| Actor update frequency | 2 |
| Target networks update frequency | 2 |
| Target networks soft-update | 0.01 |
| Target policy smoothing stddev. clip | 0.3 |
| MLP hidden size | 256 |
| Replay buffer capacity | $10^6$ |
| Discount $\gamma$ | 0.99 |
| Seed frames | 4000 |
| Exploration steps | 2000 |
| Exploration stddev. schedule | $\mathrm{linear}(1.0, 0.1, 1e6)$ |
| Action repeat | 1 |

## A.3 SEGNN

SEGNN is a recently developed neural network architecture designed to preserve the invariance and equivariance of features under 3D transformations such as rotations, translations, and reflections. It operates on an Euclidean graph which is a point cloud, where each point (node) comprises positions, node features and edge features, ensuring that the output Euclidean graph also maintains these properties. Below, we illustrate its application in the Euclidean graphs in the task Reacher_hard.

Reacher_hard: The task has rotation-equivariancy in the $xy$ plane, and no translation-invariancy due to the hinge fixed at the origin. The state contains The state contains the positions of the target, finger, hand, arm, and root (fixed at origin), and the velocities of the finger, hand, arm, and root (fixed constant $\mathbf{0}$). All these features are equivariant and they are transformed into the state-action based and state based Euclidean graphs for critic and actor, respectively. Specifically, the point set in the point cloud is $\mathcal{V} = \{\mathrm{target}, \mathrm{finger}\}$. The node feature for $\mathrm{point}_{\mathrm{finger}}$: $\mathbf{f}_{\mathrm{feature}}^{\mathrm{node,finger}} = \{[\mathbf{v}_j, \|\mathbf{v}_j\|]\}_{j \in \{\mathrm{finger,hand,arm,root}\}} \cup \{[\mathbf{x}_j, \|\mathbf{x}_j\|]\}_{j \in \{\mathrm{hand,arm,root}\}} \cup \{\mathbf{a}, \|\mathbf{a}\|\}$. The node feature $\mathbf{f}_{\mathrm{feature}}^{\mathrm{node,target}}$ for $\mathrm{point}_{\mathrm{target}}$ is $\{\mathbf{0}\}$. The node attribute contains the node_type: $\forall i \in \mathcal{V}, \mathbf{f}_{\mathrm{attribute}}^{\mathrm{node}} = [\mathrm{node\_type}(i)]$. The Observation based Euclidean graph is the same as the state-action based Euclidean graph, except that action is not included in the node feature.

Table 4: Hyperparameters for our SEGNN-based DDPG implementation for Reacher_hard.

| Hyperparameter | Setting |
|---|---|
| Learning rate | $5e{-}5$ |
| Optimizer | Adam |
| $n$-step return | 3 |
| Mini-batch size | 256 |
| Actor update frequency | 2 |
| Target networks update frequency | 2 |
| Target networks soft-update | 0.01 |
| Target policy smoothing stddev. clip | 0.3 |
| SEGNN hidden size | 64 |
| Replay buffer capacity | $10^6$ |
| Discount $\gamma$ | 0.99 |
| Seed frames | 4000 |
| Exploration steps | 2000 |
| Exploration stddev. schedule | $\mathrm{linear}(1.0, 0.1, 1e6)$ |
| Action repeat | 1 |

# B  Supplemental results

## B.1  Complete results on the effect of $\rho_{\text{aug}}$

Figure 7 presents the learning curves of all of our choices of $\rho_{\text{aug}}$ for all the 10 tasks, completing the results in Figure 5.

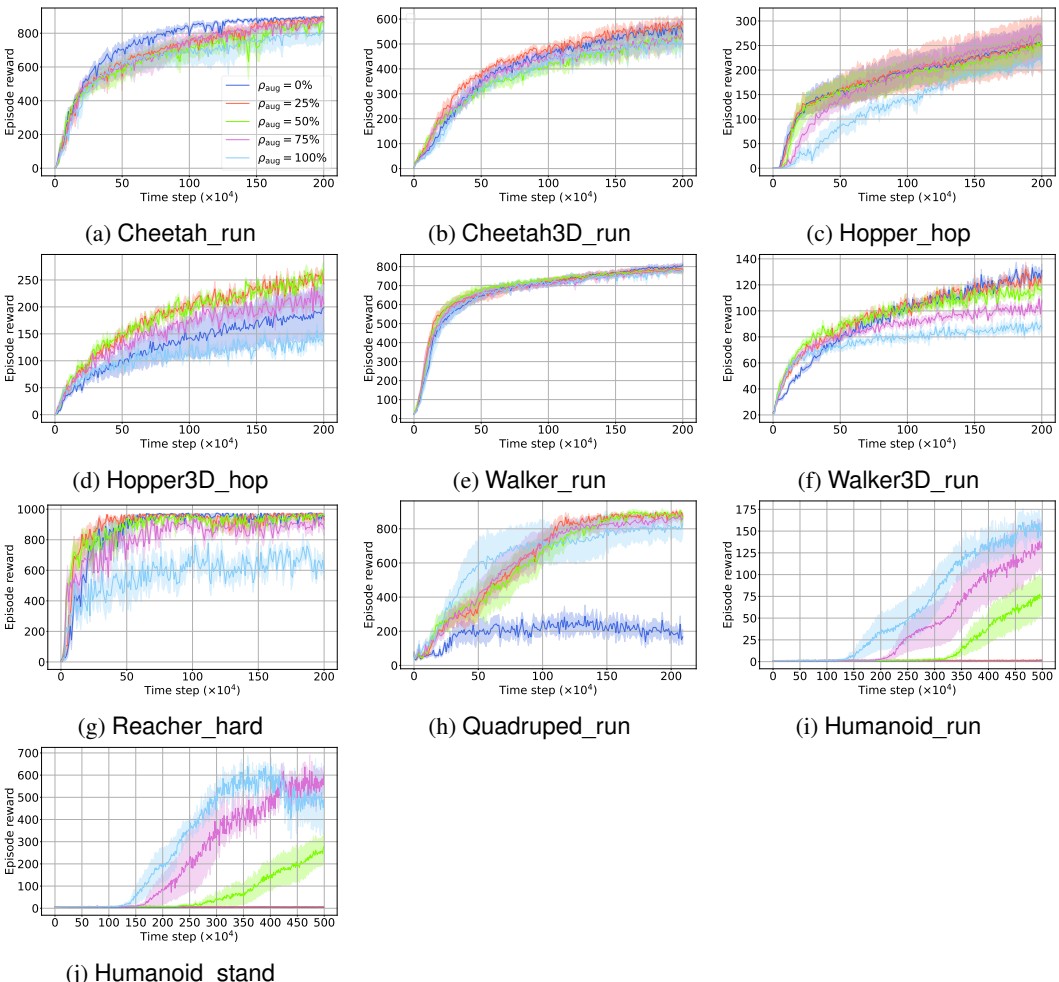

Figure 7: Learning curves of our method on the effect of $\rho_{\text{aug}}$ on all 10 tasks.

## B.2  SEGNN with single-point Euclidean graph

Figure 8 presents the results of SEGNN on Reacher_hard and Cheetah3D_run with a Euclidean graph of a single point which contains all the state features. It shows that SEGNN performs much worse than the MLP-based architectures without a good design of the graph (point set).

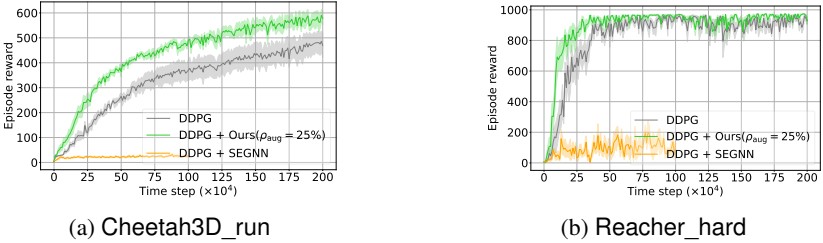

Figure 8: Learning curves of single-point SEGNN, standard DDPG, and our method.

