# OpenReview forum: "Reinforcement Learning with Euclidean Data Augmentation for State-Based Continuous Control"
_NeurIPS.cc/2024/Conference — NeurIPS 2024 poster_

### Official Review · Reviewer_5Bxo · 2024-06-23

**Soundness:** 2
**Presentation:** 3
**Contribution:** 2
**Rating:** 5
**Confidence:** 5

**Summary:**

This paper introduces a novel data augmentation method for reinforcement learning in continuous control tasks. The key innovation is leveraging Euclidean symmetries inherent in these tasks by applying rotational transformations to the original states. The authors propose using a limb-based state representation instead of the standard joint-based one to make states more amenable to these transformations. Integrated with the DDPG algorithm, this method is evaluated on 10 tasks from the DeepMind Control Suite, demonstrating improvements in data efficiency and performance, especially for complex 3D tasks. The approach outperforms existing data augmentation techniques and proves computationally efficient compared to alternative methods.

**Strengths:**

Originality:
- The idea of leveraging Euclidean symmetries for data augmentation in state-based RL is novel and underexplored.
- The limb-based state representation is an innovative solution enabling effective Euclidean augmentation.

Quality:
- Comprehensive experiments on 10 continuous control tasks, including 2D and 3D variants.
- Thorough ablation studies on the effect of augmentation ratio.
- Comparison against relevant baselines including standard RL algorithms and previous augmentation methods.

Clarity:
- The paper is well-structured.
- Limitations are discussed openly.

Significance:
- The method shows substantial improvements on challenging 3D tasks like Humanoid_run where standard methods struggle.

**Weaknesses:**

1. The paper doesn't clearly address how the method handles environments where Euclidean data augmentation might violate constraints (e.g., obstacles in AntMaze from MuJoCo).
2. The data composition process is not fully explained. Section 4.3 doesn't specify if $B_\text{aug}$ transitions are added to $B$ or substitute points in $B$.
3. The study is limited to 0-100% augmentation, while exploring larger multipliers could be insightful as one transition can be rotated with different angles
4. The focus is primarily on rotation transformations. Exploring other transformations like translation and reflection could provide a more comprehensive analysis.
5. In 5 out of 9 tasks, $ρ_\text{aug}$ = 0% achieves the best performance, suggesting the limb-based representation may be more critical than the data augmentation itself. However, the choice of limb-based representation seems motivated primarily by enabling Euclidean DA, which may limit generalizability. In addition, the advantages of limb-based over joint-based representations are not thoroughly justified.

**Questions:**

1. There appears to be a typo in line 271 - it should likely read "transform states $s_t$ and $s_{t+1}$".
2. The description of rotations in Section 4.3 is confusing. Why are there two separate rotation steps mentioned?
3. Some recent state-based data augmentation methods are not well cited in the related work, including:
     - Hindsight experience replay (Andrychowicz et al., 2017)
     - Counterfactual data augmentation (Pitis et al., 2020)
     - MoCoDA (Pitis et al., 2022)
     - Guided Data Augmentation (GuDA) (Corrado et al., 2024)
Particularly, GuDA, while focusing on offline-RL and including human guidance, describes a method for state-based data augmentation, including rotation, transformation, and reflection. Discussing these works could strengthen the paper's positioning in the field.
4. The paper focuses solely on the DDPG algorithm. Exploring compatibility with other popular RL algorithms would increase impact.

**Limitations:**

The authors adequately address limitations, noting the need for task-specific tuning and knowledge of strict Euclidean symmetries. They provide constructive suggestions for future work. Regarding societal impact, while the immediate concerns are limited for simulated tasks, a brief discussion on long-term implications for robotics, especially on real robots, could be valuable.

---

> ### Author Rebuttal · Authors · 2024-08-07
>
> Thank you for your insightful review! We are glad to provide a response to address your concerns and look forward to follow-up discussions.
>
> **On Weaknesses 1**
> Our method (Euclidean rotations) can be straightforwardly applied to constraints like obstacles: one just needs to include constraints information in the task features (see our line 186, Section 4.1).
> For example, in AntMaze, the robot is placed in a maze to navigate from a start point to an end point. Because the shape of the maze (i.e., the wall obstacles) is fixed, we can represent its shape using a directional vector and transform it (e.g., rotate) during data augmentation.
> See Figure 1(c) in our rebuttal pdf for an illustration.
>
> **On Weaknesses 2**
> We here clarify that $B_{\rm aug}$ transitions are transformed to *substitute* the original data points. This ensures that, compared to no data augmentation, we keep the same batch size and gradient update to ensure fair comparison.
> We will make this clear in our revision.
>
> **On Weaknesses 3**
> As we have clarified for Weakness 2, the upper bound of $\rho_{\rm aug}$ is 100% because we do substitution.
> Using multipliers larger than 100% needs to increase batch size, which we would avoid to ensure fair comparison with the no-augmentation baseline.
> Note that, even with substitution, a transition can still be rotated with different angles in different gradient update iterations.
>
> **On Weaknesses 4**
> One should only choose transformations that respect the symmetries in the task of interest.
> We do only rotations about the z-axis (${\rm SO}_{\vec{g}}(3)$) because it is the only valid transformation respecting the symmetries in our tasks.
> The other Euclidean transformations, i.e., translations or reflections, are not valid because:
>
> - The robots in the tasks use egocentric representation (line 256) and therefore it’s already translation-invariant.
>
> - Reflections require gait symmetry (e.g., the left leg has the same length, stiffness, etc.), which the robots in these MuJoCo tasks do not necessarily have. Some work modified the robot to enforce gait symmetry before applying reflections (e.g., see Table 1 of Corrado & Hanna labeled [1] by Reviewer X7ZX), but we did not do it.
>
> **On Weaknesses 5 - performance**
> We first clarify that we performed 10 tasks, 9 in Figure 3 and the 10th in Figure 4.
> We provide a more detailed interpretation of the results to explain the effectiveness of our method.
>
> 4 out of the 10 tasks are 2D: Figure 3’s (a)(c)(e) and Figure 4. For (a)(c)(e), $\rho_{\rm aug}$ > 0% is no better than $\rho_{\rm aug}$ = 0%; for Figure 4 (Reacher_hard), $\rho_{\rm aug}$ > 0% is better. This can be explained by the nature of the tasks:
>
> - In (a)(c)(e), a robot learns to walk/run forward with its “legs” in the 2D xz-plane, and there are no rotation symmetries within the plane itself. Our method adds the y-axis and performs rotations about the z-axis, which is valid but does not generate more data within the xz-plane that would be useful for the original 2D task. Therefore, it is expected that $\rho_{\rm aug}$ > 0% will provide limited benefit for these tasks.
>
> - In Figure 4 (Reacher_hard), a 2-limb robot is placed on the xy-plane (ground) and is tasked to rotate about the z-axis to reach a target point on the xy-plane with its tip. There are rich rotation symmetries within the 2D xy-plane, and our data augmentation generates more data therein. Therefore, it is expected that $\rho_{\rm aug}$ > 0% will be beneficial.
>
> For the other 6 tasks which are 3D, $\rho_{\rm aug}$ > 0% is slightly better than $\rho_{\rm aug}$ = 0% (especially in early training) in Figure 3’s (b)(f) and is much better in the other 4 tasks. This is because these tasks all have rotation symmetries about z-axis. Note that these tasks include the 3D version of Figure 3’s (a)(c)(e).
>
> **On Weaknesses 5 - limb-based vs joint-based representations**
> Although simulators like MuJoCo often use joint-based representations as default, both representations are convenient to obtain. Given a joint-based representation, the computation of its corresponding limb-based representation is known as *forward kinematics*, which is well-studied in robotics and implemented in simulators such as MuJoCo. We also use MuJoCo’s original APIs for forward kinematics. Therefore, choosing limb-based representations should not limit generalizability.
>
> Regarding justifying limb-based over joint-based representations, it is non-trivial to provide a theory that proves advantages of one over the other (if any), especially in the context of deep RL. That's why we have performed our experiments to give empirical evidence.
>
> **On Question 1**
> Thank you. We will fix it.
>
> **On Question 2**
> There is only one rotation applied to a chosen transition (lines 274-277). We also mention rotation in lines 271-273 just to describe how the loss is computed.
> We will modify the description to make this clear.
>
> **On Question 3**
> Thank you for suggesting the papers as they are indeed related to our work.
> As these papers are also mentioned by Reviewer X7ZX, please refer to our global response that compares our work with those papers. The comparison clarifies our different and orthogonal contributions.
>
> **On Question 4**
> Thank you for the suggestion. Our data augmentation method can be straightforwardly applied to any off-policy algorithm. We have performed additional experiments applying it to SAC on the tasks of Walker_run and Hopper3D_hop in Figure 1(a)(b) in our rebuttal pdf. The results show our data augmentation has similar effects for SAC.  These results provide evidence that our method is well compatible with other RL algorithms.
>
> **On Limitations**
> Thank you for the suggestion.
> We would not make assertive claims on real robots, since we did not provide evidence therein.
> However, as data is often even more scarce on real robots, we do believe any data augmentation method, including ours in this work, should be valued.

---

> ### Comment · Reviewer_5Bxo · 2024-08-07
>
> Thanks for the clear feedback. I think they have addressed most of my concerns.
>
> If time permits, I would still highly recommend running **"W3: The study is limited to 0-100% augmentation, while exploring larger multipliers could be insightful as one transition can be rotated with different angles."** It makes sense to ensure it is a substitution for a fair comparison with the baseline as mentioned in the response of weakness 2. However, in terms of understanding how far this method is beneficial on top of the base algorithm—given that collecting real trajectories is expensive while data augmentation is nearly zero cost—I think it would be interesting and would definitely make the paper stronger to see that for N real collected trajectories, [1, 10, 50, 100, ...] * N through augmentation on the N trajectories outperforms the base algorithm.

---

> > ### Author Response · Authors · 2024-08-08
> >
> > Thank you! We are glad our reponse have addressed your concerns.
> >
> > We are working on the suggested experiments and will update you on our progress by discussion deadline.
> > We agree that they will provide additional insights.

---

> > > ### Author Response · Authors · 2024-08-13
> > > **Updates on the suggested experiments (1/3)**
> > >
> > > Dear reviewer,
> > >
> > > We have conducted additional experiments as suggested.
> > > The primary request is to investigate whether it is beneficial to rotate transitions in a batch with different angles, as the submission only has “multipliers” up to 100%.
> > >
> > > To fulfill this request, we have run the following DDPG variants based on our limb-based data augmentation method:
> > > - Sample a batch of $B$ limb-based transitions
> > > - Perform $N(>1)$ iterations of loss computations and gradient updates
> > >     - For the first iterations, use the original, un-rotated $B$ transitions
> > >     - For each of the rest $N-1$ iterations, rotate all $B$ transitions with a random angle per transition
> > >
> > > For fair comparison, we have run similar DDPG variants of $N$ “inner-loop” iterations with the original, joint-based representation:
> > > - Sample a batch of $B$ transitions of join-based representations
> > > - Perform $N(\geq 1)$ iterations of loss computations and gradient updates using the same B transitions
> > >
> > > This way, the standard DDPG in our submission is the (B=256, N=1) case.

---

> > > > ### Author Response · Authors · 2024-08-13
> > > > **Updates on the suggested experiments (2/3)**
> > > >
> > > > Due to the time limit, we have run these DDPG variants on the tasks of Hopper3D_hop and Humanoid_run, where our method has brought significant improvements.
> > > > For Hopper3D_hop, the results in our submission have shown that a multiplier $\rho_{\rm aug}$<100% is better than $\rho_{\rm aug}$=100%.
> > > > We conjecture that a smaller batch size $B$ will favor larger multipliers, so we have run $B$=32 with $N$=1, 2, 3, 5,10 with the results shown below (we report mean and std_err over 5 seeds at 2M steps).
> > > > | N=1, No Augmentation | N=2, No Augmentation           | N=3, No Augmentation           | N=5, No Augmentation           | N=10, No Augmentation          | N=2, Our Augmentation | N=3, Our Augmentation | N=5, Our Augmentation | N=10, Our Augmentation |
> > > > |---------------------------|---------------|---------------|---------------|---------------|------------------------|------------------------|------------------------|-------------------------|
> > > > | 43.02 (17.43) | 43.14 (4.39) | 55.14 (6.25) | 42.83 (3.03) | 32.25 (8.98) | 253.14 (13.65) | 252.15 (10.06) | 250.71 (6.21) | 212.92 (7.09) |
> > > >
> > > >
> > > > For reference, the results for $B$=256 (in the submission) is shown below.
> > > > | N=1, No Augmentation (Standard DDPG) | $\rho_{\rm aug}$ = 50%, Our Augmentation |
> > > > |---------------|------------------------------|
> > > > | 27.18 (11.12) | 261.35 (15.27)               |
> > > >
> > > >
> > > > We make the following observations:
> > > > - In terms of performance, the smaller batch size of $B$=32 is qualitatively similar to $B$=256. In both cases, standard DDPG cannot learn effective policies, even with $N$>1 inner iterations, while our augmentation methods can.
> > > > - For the new experiments of data augmentation with $B$=32 and $N$>1, performance peaks at $N$=2,3 and drops with larger $N$. This suggests that using a very large $N$ is not helpful here.

---

> > > > > ### Author Response · Authors · 2024-08-13
> > > > > **Updates on the suggested experiments (3/3)**
> > > > >
> > > > > For Humanoid_run, the results in our submission have shown that a multiplier $\rho_{\rm aug}$=100% is better than $\rho_{\rm aug}$<100%, so we keep using $B$=256 and have finished running $N$=3, 5, with the results shown below (we report mean and std_err over 5 seeds at 5M steps).
> > > > > | No Augmentation, N=1 | No Augmentation, N=3 | No Augmentation, N=5 | Our Augmentation, N=3 | Our Augmentation, N=5 | Our Augmentation, $\rho_{\rm aug}$=100% |
> > > > > |----------------------|----------------------|----------------------|-----------------------|-----------------------|------------------------------|
> > > > > | 89.19 (22.16)        | 99.31 (24.65)        | 70.82 (28.99)        | 103.93 (25.61)        | 109.46 (7.15)         | 158.55 (15.13)               |
> > > > >
> > > > >
> > > > > We make the following observations:
> > > > > - Our method with the best rho_aug=100% is still better than the variants with $N$=3, 5.
> > > > > - The difference between $N$=3 and $N$=5 is not significant
> > > > >
> > > > > To summarize, we did not observe additional benefits of using multipliers larger than 100%, at least not with the straightforward variants we have experimented with.
> > > > > This might be due to the fact that, although we can generate large amounts of augmented trajectories with nearly zero cost, these “fake” trajectories are equivalent to the real ones up to a rotation, so their “marginal effects” can drop quickly after a small multiplier.

---

> > > > > > ### Comment · Reviewer_5Bxo · 2024-08-13
> > > > > >
> > > > > > Thank you for providing the new results. If I understand correctly, these results suggest that a large multiplier N does not offer significant benefits compared to the optimal $\rho_\text{aug}$ reported in the paper. My comparison focused on new results and $\rho_\text{aug}$ = 50% for Hopper3D_hop and $\rho_\text{aug}$ = 100% for Humanoid_run.
> > > > > >
> > > > > > The purpose of this comparison was to understand the relative contributions of proper representation (i.e., limb-based) versus Euclidean augmentation to performance improvement. Combining the new results with Figures 3 and 4 from the original paper, it appears that:
> > > > > > 1. In 7 out of 10 environments (Cheetah_run, Cheetah3D_run, Hopper_hop, Hopper3D_hop, Walker_run, Walker3D_run, Reacher_Hard), DDPG+Ours ($\rho_\text{aug}$ = 0%) achieves near-optimal performance. This suggests that the change in representation alone is beneficial.
> > > > > >
> > > > > > 2. In 6 out of 10 environments (Cheetah_run, Cheetah3D_run, Hopper_hop, Walker_run, Quadruped_run, Reacher_hard), standard DDPG achieves near-optimal performance. In addition, the DDPG performance reported in this paper appears to be lower than that reported in the DeepMind Control Suite paper [1], Figure 5.
> > > > > >
> > > > > > These observations raise concerns about the method's effectiveness: The performance improvement appears marginal and requires careful selection of the $\rho$ value for each task. Increasing the amount of data does not seem to provide additional benefits, which undermines the motivation for using this as a data augmentation method. It is not guaranteed to outperform default DDPG without changes to the representation configuration.
> > > > > >
> > > > > > I do appreciate authors' efforts on addressing my questions, but given these concerns, I maintain my original assessment and score.

---

> > > > > > > ### Author Response · Authors · 2024-08-14
> > > > > > >
> > > > > > > Dear reviewer,
> > > > > > >
> > > > > > > Thank you and we appreciate your updated assessment!
> > > > > > >
> > > > > > > We make two additional comments based on your feedback.
> > > > > > >
> > > > > > > 1. We would like to emphasize that we did not cherry pick tasks where our method outperforms others; instead, we sampled a range of tasks with various degrees of symmetries to show when and why our method can bring benefits.
> > > > > > > Our results show clear advantages of our method on tasks that exhibit rotation symmetries about gravity (e.g., 2D Reacher_hard, most 3D tasks), where doing *both* limb-based representation and a $\rho$ >0% on top of it is crucial to achieve the best performance.
> > > > > > > For other 2D tasks, it is expected that $\rho$ >0% will bring limited benefits over $\rho$ =0%, as we have explained. We plan to move these experiments to appendix in our revision, leaving room to incorporate the discussion and new results during rebuttal.
> > > > > > >
> > > > > > > 2. It is pretty much expected that the performance will saturate as multipliers $\rho$ or $N$ gets larger and that the optimal multiplier is task-dependent, which is also observed in prior work, including the very recent Corrado & Hanna, ICLR 2024.
> > > > > > > This issue holds true equally for our novel data transformation and for other transformations.
> > > > > > > As discussed in Limitations, we do not claim contributions on tuning this multiplier, which should be a separate contribution that would benefit our augmentation method as well as others.

---

> > > > > > > > ### Comment · Reviewer_5Bxo · 2024-08-14
> > > > > > > >
> > > > > > > > Thank you for your responses. I still maintain my opinion about the limited effectiveness of the method, given that:
> > > > > > > >
> > > > > > > > 1. The performance boost requires some specific task properties, i.e., rotation symmetries about gravity.
> > > > > > > >
> > > > > > > > 2. Even for tasks where the method demonstrates improvement, it requires:
> > > > > > > > a) Changes in configurations
> > > > > > > > b) Additional hyperparameter tuning
> > > > > > > >
> > > > > > > > 3. I still question the difference between the DDPG performance and the performance reported in [1] Figure 5. This might lead to marginal performance differences between the proposed method and the default DDPG setting if we choose hyperparameters carefully.
> > > > > > > >
> > > > > > > > However, I appreciate the authors' additional experiments and final response, which, based on the paper, do help us better understand the method. Therefore, I have increased my score accordingly.
> > > > > > > >
> > > > > > > > [1] Tassa, Yuval, et al. "Deepmind control suite." arXiv preprint arXiv:1801.00690 (2018).

---

> > > > > > > > > ### Author Response · Authors · 2024-08-14
> > > > > > > > >
> > > > > > > > > Thanks.
> > > > > > > > >
> > > > > > > > > As the discussion period comes to its end, we authors thank you for your reviewing!

---

> > > > > > > > > > ### Comment · Reviewer_5Bxo · 2024-08-14
> > > > > > > > > >
> > > > > > > > > > Also appreciate your efforts. Best wishes!

---

### Official Review · Reviewer_rWGW · 2024-06-30

**Soundness:** 3
**Presentation:** 2
**Contribution:** 2
**Rating:** 5
**Confidence:** 2

**Summary:**

This paper introduces a novel data augmentation strategy tailored for reinforcement learning (RL) agents operating in state-based continuous control environments. The method leverages limb-based state features rather than joint-based configurations, allowing for more effective augmentation. Experiments conducted on various tasks from the DeepMind Control Suite demonstrate significant improvements in data efficiency and performance over standard RL algorithms and existing augmentation methods.

**Strengths:**

The paper introduces a unique approach to data augmentation for RL in state-based continuous control, moving away from traditional perturbation methods to Euclidean transformations. This approach is innovative and addresses the limitations of existing methods.

**Weaknesses:**

Firstly, I am not an expert in this research field. I think this paper lacks some deeper theoretical support. The scenario covered by Theorem 1 is overly simplistic. The proposed method involves additional computations for applying Euclidean transformations and managing the limb-based state features. The paper could benefit from a more detailed discussion on the computational overhead and potential optimizations.

**Questions:**

Could the authors provide more details on how to tune the various parameters involved in the proposed method, such as the choice of Euclidean transformations and the proportion of augmented data?

**Limitations:**

The authors acknowledge the need for task-specific tuning of the hyperparameter and the requirement for knowledge of strict Euclidean symmetries. They have adequately addressed the limitations.

---

> ### Author Rebuttal · Authors · 2024-08-07
>
> Thank you for your insightful review! We are glad to provide a response and look forward to follow-up discussions.
>
> **On Weaknesses - theoretical support**
> The paper does not state a Theorem 1. Can you further clarify your concern? Thanks.
>
> **On Weaknesses - computational overhead**
> Our method requires minimal additional computation. The only additional computation is transforming a subset of transitions in the mini-batch, as described in Section 4.3. Note that, compared to the standard method with no data augmentation, we keep the same batch size and gradient update to ensure no additional computation is required. The transformation is essentially matrix multiplication (involving 3D rotation matrices), which incurs negligible overhead in the training process, as we have shown in Figure 4 (bottom).
>
> **On Questions - choices of Euclidean transformations**
> The available Euclidean transformations are not tunable, because one should only choose transformations that respect the symmetries in the task of interest.
> We focus on rotation symmetries about the z-axis (${\rm SO}_{\vec{g}}(3)$) because it is the only available symmetries in our tasks.
> The other Euclidean transformations, i.e., translations or reflections, are not available because:
>
> - The robots in the tasks use egocentric representation (line 256) and therefore it’s already translation-invariant.
>
> - Reflections require gait symmetry (e.g., the left leg has the same length, stiffness, etc.), which the robots in the tasks do not necessarily have. Some prior work modified the robot to enforce gait symmetry before applying reflections (e.g., see Table 1 of Corrado & Hanna labeled as [1] by Reviewer X7ZX), but we did not do it.
>
> **On Questions - tuning hyperparameters**
> $\rho_{\rm aug}$ is our only additional hyperparameter on top of an off-the-shelf algorithm like DDPG. Its tuning is standard, no different than other hyperparameters:
>
> - We did grid search, which is standard and the same way as tuning other DDPG hyperparameters
>
> - We have discussed the effect of $\rho_{\rm aug}$ in Section 5.3, which gives insights on tuning strategy. In short, tasks with richer symmetries benefit from a larger $\rho_{\rm aug}$.
>
> - An automatic tuning method might be beneficial, which we leave for future work (Section 6).

---

> > ### Comment · Reviewer_rWGW · 2024-08-13
> > **Thank you**
> >
> > Thank you for the response. Since I am not an expert in this field, I will maintain my current neutral score. Thank you again or your detailed responses to my review.

---

> > > ### Author Response · Authors · 2024-08-13
> > >
> > > Thank you!

---

### Official Review · Reviewer_X7ZX · 2024-07-04

**Soundness:** 2
**Presentation:** 2
**Contribution:** 1
**Rating:** 4
**Confidence:** 4

**Summary:**

This paper proposes a data augmentation technique that leverages Euclidean symmetries (e.g. rotational symmetry) in a task's dynamics to generate augmented data. When a task's state features do not have such symmetries, the paper also discusses how to define a new state representation with these symmetries (so that data augmentation can be applied). Empirically, the paper shows that RL is more data efficient with this data augmentation technique than without it on various DMControl tasks.

**Strengths:**

1. The paper focuses on a class of augmentations that have been under-studied (compared to visual augmentations)
2. Redefining a task's state representation to enable data augmentation is an interesting concept to me.
3. Comparing the proposed augmentation strategy with an RL agent that learns with an equivariant network architecture was useful; I've always wondered how these two different approaches compared.

**Weaknesses:**

1. **The story seems improperly situated in the data augmentation literature.** While it is true that most prior data augmentation works focus on perturbation-based augmentations, many prior works have exploited task symmetries and invariances to generate additional data that agrees with the task's dynamics and reward function [1-6]. Corrado et. al [1] calls these dynamics-invariant augmentations.

2. **It’s unclear if this change of representation is necessary for data augmentation.** Corrado et. al [1-2] perform data augmentation using SO(3) rotations in MuJoCo tasks similar to those studied in this paper *without* changing the task’s state features.

3. **Weak empirical results.** In 5/9 tasks, DDPG + data augmentation performs just as well as DDPG without data augmentation. With 5 seeds per curve, I have low confidence in the observed benefits in the remaining tasks; RL is notoriously high variance, and the variance between runs is enough to create statistically different distributions. 95% confidence belts assume the distributions of returns at each evaluation point are normally distributed, which is likely not the case.

[1] Corrado & Hanna. "Understanding when Dynamics-Invariant Data Augmentations Benefit Model-free Reinforcement Learning Updates." ICLR 2024.

[2] Corrado et. al. "Guided Data Augmentation for Offline Reinforcement Learning and Imitation Learning." arXiv:2310.18247

[3] Pitis et. al. “Counterfactual Data Augmentation using Locally Factored Dynamics.” NeurIPS 2020.

[4] Pavlov et. al. "Run, Skeleton, Run: Skeletal Model in a Physics-Based Simulation." AAAI 2018.

[5] Abdolhosseini et. al. “On Learning Symmetric Locomotion.” ACM SIGGRAPH 2019.

[6] Adrychowicz et. al. "Hindsight Experience Replay." NeurIPS 2017.

[7] Henderson et. al. "Deep Reinforcement Learning that Matters." AAAI 2018

**Minor Comments**
1. I think section 3.2 can be omitted; the augmentation framework described in section 4.3 can be applied to any off-policy RL algorithm.

6. Line 271: typo, I think the second $s_t$ should be $s_t\prime$

**Questions:**

1. Line 40-43: Could the authors elaborate on what “uncorrelated” means here?

4. Line 159: Does anything change if the morphology tree contains a cycle?

5. Line 272: What does it mean to keep $a_t$ and $r_t$ invariant?

**Limitations:**

The paper addresses limitations, though I think an additional limitation should be emphasized: not only may itbee difficult to specify a symmetry, symmetries might not even exist in some tasks.

I don't believe this limitation takes anything away from the paper though; it's a core limitation of most non-perturbation-based data augmentation methods, and it's fine if these methods only apply to some tasks. That's simply the nature of data augmentation. Thankfully, in many real-world tasks (especially robotics tasks where you have an agent acting in 3D space) a human *can* identify symmetries. Pitis et. al [1] and Corrado et. al [2] discuss this point too.

[1] Pitis et. al. “Counterfactual Data Augmentation using Locally Factored Dynamics.” NeurIPS 2020.

[2] Corrado & Hanna. "Understanding when Dynamics-Invariant Data Augmentations Benefit Model-free Reinforcement Learning Updates." ICLR 2024.

---

> ### Author Rebuttal · Authors · 2024-08-07
>
> Thank you for your insightful review! We are happy to provide a response. We hope it can trigger your re-evaluation and look forward to follow-up discussions.
>
> **On Weakness 1**
> Thank you for suggesting the papers as they are indeed related to our work.
> As some of the papers are also mentioned by Reviewer 5Bxo, please refer to our global response that compares our work with those papers. The comparison clarifies our different and orthogonal contributions.
>
>  **On Weakness 2**
> Indeed, one can perform SO(3) rotations on the original joint-based state representation. In our tasks, only the torso ($i=1$) has features for orientation (in its up to 6 DoFs) while features of other limbs ($i>1$) are just angles and angular velocities of the hinges. Therefore, under a rotation, only the torso’s orientation features are changed, while other features stay unchanged. This is what prior work such as Corrado et. al. did.
>
> However, we hypothesized this data augmentation by rotating original joint-based features would bring little benefit to locomotion tasks. This is because torso features make up only a fraction of all features when the number of limbs is large (which is the case in our tasks). Our limb-based representation instead rotates all limbs to provide richer augmentation.
>
> Our hypothesis is supported by results of Corrado & Hanna [1] (see Figure 16 of [1]: confidence intervals are overlapped for Ant; rotation underperforms than no-augmentation for Humanoid).
> This is also evidenced by our added results in Figure 3 in the rebuttal pdf: we have added curves corresponding to performing rotation data augmentation on original state features, which is less performant than ours.
>
>  **On Weakness 3 - effectiveness**
> We first clarify that we have performed 10 tasks, 9 in Figure 3 and the 10th in Figure 4.
> We here provide a more detailed interpretation of the results to explain the effectiveness of our method.
>
> 4 out of the 10 tasks are 2D: Figure 3’s (a)(c)(e) and Figure 4. For (a)(c)(e), $\rho_{\rm aug}$ > 0% is no better than $\rho_{\rm aug}$ = 0%; for Figure 4 (Reacher_hard), $\rho_{\rm aug}$ > 0% is better. This can be explained by the nature of the tasks:
>
> - In (a)(c)(e), a robot learns to walk/run forward with its “legs” in the 2D xz-plane, and there are no rotation symmetries within the plane itself. Our method adds the y-axis and performs rotations about the z-axis, which is valid but does not generate more data within the xz-plane that would be useful for the original 2D task. Therefore, it is expected that $\rho_{\rm aug}$ > 0% will provide limited benefit for these tasks.
>
> - In Figure 4 (Reacher_hard), a 2-limb robot is placed on the xy-plane (ground) and is tasked to rotate about the z-axis to reach a target point on the xy-plane with its tip. There are rich rotation symmetries within the 2D xy-plane, and our data augmentation generates more data therein. Therefore, it is expected that $\rho_{\rm aug}$ > 0% will be beneficial.
>
> For the other 6 tasks which are 3D, $\rho_{\rm aug}$ > 0% is slightly better than $\rho_{\rm aug}$ = 0% (especially in early training) in Figure 3’s (b)(f) and is much better in the other 4 tasks. This is because these tasks all have rotation symmetries about z-axis, so our data augmentation is beneficial. Note that these tasks include the 3D version of Figure 3’s (a)(c)(e).
>
> To summarize, we did not cherry pick tasks where our method outperforms others; instead, we sampled a range of tasks with various degrees of symmetries to show when and why our method can bring benefits.
>
>  **On Weakness 3 - seeds and confidence intervals**
> For the 4 tasks where $\rho_{\rm aug}$ > 0% is better, we have completed another 5 seeds during rebuttal.
> With the 10 seeds,  Figure 2 in the rebuttal pdf updates the training curves with mean+95% CI.
> Figure 2 also gives inter-quartile means (IQMs) with 95% bootstrap confidence intervals, which is more robust to outliers  (used in prior work like Corrado et. al. [2]).
> We believe these results give enough confidence on the effectiveness of our method.
>
>  **On Minor Comment 1**
> Thank you for the suggestion. We will consider removing Section 3.2 in our revision.
>
>  **On Minor Comment 2**
> Thank you. We will fix it.
>
>  **On Question 1**
> We mean, although the original transition comes from the ground truth dynamics and reward functions, the perturbed transition does not necessarily. In this sense, they are “uncorrelated”. In the words of Corrado et. al., perturbation is not a Dynamics-Invariant data augmentation.
>
>  **On Question 2**
>  For morphology trees containing cycles, our key idea and methodology, i.e., ${\rm SO}_{\vec{g}}(3)$ data augmentation on limb-based representation, still applies well. One just needs to identify the kinematic features therein and perform appropriate transformations.
> We focus on no-cycle trees because most locomotion tasks (including all of our tasks) do not contain a cycle, and the torso/base often serves as the tree root. In our tasks, the root is special in the sense that it can have up to 6 DoFs, while other nodes only have hinge-like DoFs. Again, we focus on this case to make our method description clear, but our methodology applies to cycles.
>
>  **On Question 3**
> That simply means we do not change them.
> Since actions a_t are scalar torques, they should not change under rotations.
> Rewards should also stay the same under rotations.

---

> > ### Author Response · Authors · 2024-08-13
> >
> > Dear reviewer,
> >
> > As the discussion deadline approaches, we would like to know if our response has addressed your concerns. Should any concerns remain, we will gladly address them.
> >
> > Thank you again for reviewing our paper.

---

> > ### Comment · Reviewer_X7ZX · 2024-08-13
> >
> > Thank you for the detailed response! I'm glad to see the comparisons with the prior works; including a some discussion on them would better situate this paper in the literature. It would also emphasizes some of the novelty by clarifying how these augmentations differ from those in prior works, particularly this part of your response:
> >
> > > However, we hypothesized this data augmentation by rotating original joint-based features would bring little benefit to locomotion tasks. This is because torso features make up only a fraction of all features when the number of limbs is large (which is the case in our tasks). Our limb-based representation instead rotates all limbs to provide richer augmentation.
> >
> > Regarding experiments: It would be clearer just have the 3D tasks where augmentation helps in the main paper. You can explain that these augmentations would not be helpful in the 2D tasks, and then point the reader to an appendix containing the 2D experiments. I have raised my score because of these clarifications.
> >
> > All of my questions have been answered. Just a quick follow-up comment: I think a clearer alternative to "uncorrelated" would be to say something like "the augmented data generally does not agree with the task's dynamics" or "the augmented data is generally not dynamics-invariant."

---

> > > ### Author Response · Authors · 2024-08-13
> > >
> > > Thank you!
> > > We will adopt your suggestion on "uncorrelated".

---

### Official Review · Reviewer_aR8o · 2024-07-11

**Soundness:** 3
**Presentation:** 3
**Contribution:** 3
**Rating:** 5
**Confidence:** 4

**Summary:**

This work proposes a novel data augmentation approach that leverages Euclidean symmetry for continuous control to improve the efficiency and performance of RL algorithms. The authors integrated their approach with DDPG and performed a series of comparison against the vanilla SAC algorithm, other augmentation techniques, and other equivariant methods. The results show some improvements in performance across a range of continuous control tasks, especially those with rich 3D motions and large number of limbs and joints.

**Strengths:**

1. The proposed approach improved the learning efficiency of RL algorithms (demonstrated with DDPG in this work) without the need of changes to the algorithm, which lowers the barrier for adaptability for the community.
2. Euclidean transformations maintain the inherent physics of the task, ensuring that the augmented data are still representative of realistic scenarios. This preservation is crucial for the relevance and usefulness of the augmented data.

**Weaknesses:**

1. This method is primarily applicable to tasks with clear Euclidean symmetries. So, it seems that in environments where such symmetries are not evident or relevant, the approach may not be applicable or effective. While the 'Limitation' section briefly touches upon this issue, it is not clear what categories of continuous control problems this approach will be applicable to.
2. The effectiveness of Euclidean transformations may vary depending on whether the task is set in a 2D or 3D space, with potentially limited benefits in more constrained settings (like planar movements).
3. As the 'Limitation' section suggests, practical application of the framework will be difficult due to the task-specific hyperparameter tuning requirements. While the author suggests adaptive hyperparameter tuning as a potential future work, some insights into what characteristics of the environment dynamics dictate the optimal choice of hyperparameters such as \rho_{aug} would be helpful. Also, in the context of practical application, switching from joint-based to limb-based configurations complicate the state representation and requires significant changes to the training environment.

**Questions:**

1. In Section 5.1, the authors say "To summarize, our method reliably improves DDPG and SAC, the state-of-the-art RL algorithms on the continuous control tasks." However, it looks like only vanilla SAC is used in this work for comparison. Hence, the mention of improving SAC should be removed.

2. In line 194 p^y_1 should be  p^z_1.

**Limitations:**

Technical limitations are highlighted in the paper.

---

> ### Author Rebuttal · Authors · 2024-08-07
>
> Thank you for your insightful review! We are glad to provide a response and look forward to follow-up discussions.
>
> **On Weakness 1**
> Our work focuses on robot locomotion tasks as instances of continuous control, which clearly exhibit Euclidean symmetries. Other robotics tasks (e.g., navigation) and many applications that operate in the 3D physical space should also exhibit Euclidean symmetries.
> We agree that there are continuous control tasks that might not exhibit Euclidean symmetries, such as those in electrical and power engineering.
> We will update our "Limitation" section and other parts of the paper to clarify this.
>
> **On Weakness 2**
> We agree that the effectiveness of Euclidean transformations depends on the task, and our results indicate that it is even more subtle than just 2D vs 3D, for which we provide a more detailed interpretation below. In short, Euclidean transformations are effective if the task itself exhibits corresponding symmetries; otherwise, one should not expect these transformations to improve the performance in the first place.
>
> Our experiments include 10 tasks, 9 in Figure 3 and the 10th in Figure 4.
>
> 4 out of the 10 tasks are 2D: Figure 3’s (a)(c)(e) and Figure 4. For (a)(c)(e), $\rho_{\rm aug}$ > 0% is no better than rho_aug = 0%; for Figure 4 (Reacher_hard), $\rho_{\rm aug}$ > 0% is better. This can be explained by the nature of the tasks:
>
> - In (a)(c)(e), a robot learns to walk/run forward with its “legs” in the 2D xz-plane, and there are no rotation symmetries within the plane itself. Our method adds the y-axis and performs rotations about the z-axis, which is valid but does not generate more data within the xz-plane that would be useful for the original 2D task. Therefore, it is expected that $\rho_{\rm aug}$ > 0% will provide limited benefit for these tasks.
>
> - In Figure 4 (Reacher_hard), a 2-limb robot is placed on the xy-plane (ground) and is tasked to rotate about the z-axis to reach a target point on the xy-plane with its tip. There are rich rotation symmetries within the 2D xy-plane, and our data augmentation generates more data therein. Therefore, it is expected that $\rho_{\rm aug}$ > 0% will be beneficial.
>
> For the other 6 tasks which are 3D, $\rho_{\rm aug}$ > 0% is slightly better than $\rho_{\rm aug}$ = 0% (especially in early training) in Figure 3’s (b)(f) and is much better in the other 4 tasks. This is because these tasks all have rotation symmetries about z-axis, so our data augmentation is beneficial. Note that these tasks include the 3D version of Figure 3’s (a)(c)(e).
>
> **On Weakness 3 - tuning $\rho_{\rm aug}$**
> Per our response to Weakness 2, our results have provided the following insights:
> Tasks with richer symmetries often benefit from larger values of $\rho_{\rm aug}$ (e.g., $\rho_{\rm aug}$=100% is best for Humanoid tasks).
>
> **On Weakness 3 - limb-based vs joint-based configurations**
> Although simulators like MuJoCo often use joint-based configurations as default, both configurations are convenient to obtain. Given a joint-based configuration, the computation of its corresponding limb-based configuration is known as “forward kinematics”, which is well-studied in robotics and implemented in simulators such as MuJoCo. We did use MuJoCo’s original APIs for forward kinematics. Therefore, we argue using limb-based configurations is not a significant issue in practice.
>
> **On Question 1**
> Thank you for the suggestion. We will remove "SAC" in the statement.
>
> **On Question 2**
> Thank you. We will fix it.

---

> > ### Author Response · Authors · 2024-08-13
> >
> > Dear reviewer,
> >
> > As the discussion deadline approaches, we would like to know if our response has addressed your concerns. Should any concerns remain, we will gladly address them.
> >
> > Thank you again for reviewing our paper.

---

> > ### Comment · Reviewer_aR8o · 2024-08-13
> > **Response to rebuttal**
> >
> > Thanks for the responses to my questions. I am sticking to my score.

---

> > > ### Author Response · Authors · 2024-08-13
> > >
> > > Thank you!

---

### Author Rebuttal · Authors · 2024-08-07

We attach here a pdf that contains additional results and illustrations, which we refer to in our response to individual reviews.

We below compare our work with papers [1-6] suggested by reviwer X7ZX to clarifies our different and orthogonal contribution.
Some of these papers are also suggested by reviewer 5Bxo.

**Our contribution**
Our contribution is a novel data augmentation transformation, namely ${\rm SO}_{\vec{g}}(3)$ transformation on limb-based state representations, for RL-based robot locomotion. Our experiments on a wide range of simulated locomotion tasks show the advantage of this transformation over alternatives in sample and computation efficiency.

**Comparision with [1-6]**
Our contribution is different from the papers [1-6] in the following sense:

- [1,2] and our work both consider “dynamics-invariant” data augmentation transformations, but our focus is very different. [1,2] focus on better leveraging known, existing dynamics-invariant transformations, drawing their conclusions mostly from robot navigation and manipulation tasks (e.g., AntMaze, Soccer, panda-gym); while we propose a novel transformation for robot locomotion.

    - Specifically, [1] studies the question of when data augmentation is helpful and concludes the augmentation ratio is crucial (their contribution 3). Their paper’s main body focuses on robot navigation and manipulation, where transformations like goal-relabelling and translations are key to overcoming the sparse reward issue. [1] does use rotation too, in a toy 2D navigation task (Goal2D-v0) where the robot is a particle (no meaningful kinematics) and in the dense-reward MuJoCo locomotion tasks (their Appendix F) where the rotation is performed on joint-based representation and does not yield improvement over no-augmentation baseline (see our response to Weakness 2 for more details).

    - [2] uses transformations to randomly generate synthetic data and then asks humans to select high-quality ones. Such manual filtering is helpful when doing robot navigation, which [2] focuses on, because humans can easily tell high-quality trajectories based on distance to goal; but it is much more difficult for humans to tell effective joint movements for locomotion. Our method does not require human effort at all.

- [3] (and their follow-up work MoCoDA [Pitis et al., 2022]) proposes a data augmentation transformation that requires local (causal) independence, so that augmentation can be performed via stitching independent trajectories from decomposed, independent parts, which is useful for tasks like particles moving and 2-arm robot with static base. We focus on locomotion tasks that do not exhibit sparse kinematic interactions between limbs and therefore cannot benefit much from [3]’s method. For example, the cheetah’s two legs are connected through the moveable torso, and therefore we cannot decompose and stitch their separate trajectories.

- [4] focuses on the transformation of reflection that exploits bilateral gait symmetry. Reflections require gait symmetry (e.g., the left leg has the same length, stiffness, etc.), which our MuJoCo tasks do not necessarily have. Some work modifies the MuJoCo robot to enforce gait symmetry before applying reflections (e.g., see Table 1 of [1]). Our rotation transformation instead does not require gait symmetry.

- [5] also focuses on reflections (they call them "mirror") for locomotion. They perform trajectory-level transformations for on-policy RL, which is technically not sound but they found it "not a critical issue in practice", while we do rotation-based transition-level transformations for off-policy RL. Moreover, their data augmentation does not yield significant improvement over the no-augmentation baseline (see their Figure 3, DUP vs BASE).

- [6] addresses the problem of sparse rewards in reinforcement learning by the goal-relabelling transformation, which is different from our rotation transformation and does not bring much benefit to locomotion tasks.

---

### Decision · Program_Chairs · 2024-09-25

**Decision:**

Accept (poster)

**Comment:**

This paper introduces a novel data augmentation technique for reinforcement learning in continuous control tasks, leveraging euclidean symmetries. The primary contributions include a method to apply Euclidean transformations to state representations, a limb-based state representation enabling effective augmentation, and empirical evaluation on various DeepMind Control Suite tasks.

The reviewers generally agree on the strengths: a novel approach to state-based RL data augmentation, comprehensive experiments, and significant improvements on challenging 3D tasks. However, concerns were raised regarding limited theoretical support, varying effectiveness across tasks, and focus primarily on rotational transformations. There was some disagreement among reviewers regarding the significance of the contribution relative to prior work on dynamics-invariant augmentations and the overall effectiveness of the method given its varied performance across tasks.

Despite these limitations, the paper presents a promising approach with some notable results on complex tasks. The authors have adequately addressed most concerns during the discussion phase. Therefore, I recommend accepting this paper for a poster presentation.